# Stress Response Analysis via Dynamic Entropy in EEG: Caregivers in View

**DOI:** 10.3390/ijerph20105913

**Published:** 2023-05-22

**Authors:** Ricardo Zavala-Yoé, Hafiz M. N. Iqbal, Roberto Parra-Saldívar, Ricardo A. Ramírez-Mendoza

**Affiliations:** 1Tecnológico de Monterrey, Calzada del Puente, 222. Col. Ejidos de Huipulco, Mexico City 14380, Mexico; 2Tecnológico de Monterrey, Eugenio Garza Sada 2501, Monterrey 64849, Mexico

**Keywords:** time series dynamic entropy, stress quantification, brain dynamics

## Abstract

According to the World Health Organization (WHO), stress can be defined as any type of alteration that causes physical, emotional, or psychological tension. A very important concept that is sometimes confused with stress is anxiety. The difference between stress and anxiety is that stress usually has an existing cause. Once that activator has passed, stress typically eases. In this respect, according to the American Psychiatric Association, anxiety is a normal response to stress and can even be advantageous in some circumstances. By contrast, anxiety disorders differ from temporary feelings of anxiousness or nervousness with more intense feelings of fear or anxiety. The Diagnostic and Statistical Manual (DSM-5) explicitly describes anxiety as exorbitant concern and fearful expectations, occurring on most days for at least 6 months, about a series of events. Stress can be measured by some standardized questionnaires; however, these resources are characterized by some major disadvantages, the main one being the time consumed to interpret them; i.e., qualitative information must be transformed to quantitative data. Conversely, a physiological recourse has the advantage that it provides quantitative spatiotemporal information directly from brain areas and it processes data faster than qualitative supplies. A typical option for this is an electroencephalographic record (EEG). We propose, as a novelty, the application of time series (TS) entropies developed by us to inspect collections of EEGs obtained during stress situations. We investigated this database related to 23 persons, with 1920 samples (15 s) captured in 14 channels for 12 stressful events. Our parameters reflected that out of 12 events, event 2 *(Family/financial instability/maltreatment)* and 10 *(Fear of disease and missing an important event)* created more tension than the others. In addition, the most active lobes reflected by the EEG channels were frontal and temporal. The former is in charge of performing higher functions, self-control, self monitoring, and the latter is in charge of auditory processing, but also emotional handling. Thus, events E2 and E10 triggering frontal and temporal channels revealed the actual state of participants under stressful situations. The coefficient of variation revealed that E7 *(Fear of getting cheated/losing someone)* and E11 *(Fear of suffering a serious illness)* were the events with more changes among participants. In the same sense, AF4, FC5, and F7 (mainly frontal lobe channels) were the most irregular on average for all participants. In summary, by means of dynamic entropy analysis, the goal is to process the EEG dataset in order to elucidate which event and brain regions are key for all participants. The latter will allow us to easily determine which was the most stressful and on which brain zone. This study can be applied to other caregivers datasets. All this is a novelty.

## 1. Introduction

The WHO defines stress as any type of alteration that causes physical, emotional, or psychological tension [1]. A natural response to stress is anxiety. In such an action–reaction definition, since stress is the input, it is usually generated by an actual cause (the stressor), contrary to anxiety [2]. The Diagnostic and Statistical Manual (DSM-5) [3] explicitly describes anxiety as exorbitant concern and fearful expectations, occurring on most days for at least 6 months, about a series of events. Its bodily manifestation is muscle tenseness and the psychological exhibition is evasion comportment. As a consequence of these definitions, the need to quantify this condition arises.

In [4], it is explained how to measure and distinguish psychological and physiological stress. In addition, organic causes and consequences are also explored to finally offer a table-guide to choose appropriate stress measures. Afterwards, a short questionnaire is designed with essential questions for following best practices in choosing an appropriate stress-related biomarker. Here, a biomarker-stress model structure is proposed which suggests that a stressful event, X, leads to a biological change, Y, that then leads to the disease state or related outcome, Z, but we acknowledge that including the organic component is quantitatively difficult. In this study, psychological standard questionnaires are not considered.

With the focus on including the psychological exploration in quantifying stress and anxiety, a typical process is applying standard psychological questionnaires with numerical outcomes after interpretation. These quantities reflect a degree of this alteration. Further work here defines a linear (or logistic) regression model to find a mathematical explanation of the linguistic data [5,6]. Therefore, a complete psychological–mathematical construction is presented. However, despite including psychological information, the cost of this is that such qualitative data must be transformed to quantitative resources by means of interpretation and scales association. At this point, a more objective mean is required. A frequent solution is using an electroencephalographic record. The advantage of this is that it gives time–space information directly from brain areas. Further, an EEG interpretation is performed faster than its qualitative counterpart [7]. Indeed, using EEG opens a lot of math modeling possibilities because an EEG record is a collection of time series of sampled voltages. These arguments are familiar because the binomial (EEG, stress) appears.

Therefore, we realize that stress and anxiety are essential issues that can be objectively identified by EEG signals. Traditionally, EEG are applied to patients or subjects to detect alterations that may cause or trigger serious illnesses. For example, in [8,9,10,11,12], a family of entropy parameters was applied to EEG collections to reveal features that were not evident in severe epileptic syndromes. In this sequence of studies (and many others), the main actors are the affected patients. However, even when the main protagonists were children, quietly, next to them, there also exist a key group of people that are frequently understudied from this tension perspective: the ***caregiver***. Caregivers’ necessities are regularly undervalued. In this sense, unlike other serious chronic medical conditions, the impact of epilepsy on the family and caregivers establishes a particularly understudied area [13,14,15,16]. Many efforts have been made to investigate stress in caregivers of many diseases, but stress in caregivers of patients with epilepsy is even less investigated. In [5,6,14,15], the burden of these caretakers is evaluated but with the support of psychological questionnaires and without EEG databases.

As far as we are concerned, the availability of electroencephalograms for caregivers is extremely scarce or missing. As a result of this, investigations that come from direct EEG processing are also sparse. This event may be a consequence of the fact that subjecting a participant to a stressor is controversial from an ethical perspective. For instance, in [17], a set of EEGs are processed to obtain conclusions about sleep and fatigue in female caregivers of elderly people and questionnaires are not used. Although the database is well described, it is not available online. Conversely, in [7], a family of 20 papers that report results about stress from EEG is investigated. Nevertheless, instead of putting participants through stressors, a “smoothed” version of them was used. In all the investigations, subjects were asked to answer questions about basic mathematics, contradictions by simultaneous color–word stimuli, games, memory exercises, etc. This report described the math analysis performed in those 20 documents. Basically, they work in time and frequency domain. In addition, they use linear and non-linear statistics. Among the latter, a member of the time series entropies is utilized (spectral entropy). On the other hand, another type of entropy plus machine learning is applied to detect distress but in affective situations in [18]. However, in [19], visually elicited emotional states of calmness versus negative stress of non-caregivers is contrasted. A huge database containing information on cognitive status and personality measures is described in [20] and, although data are available online, stress and, even more, the stress of custodians is not considered.

Applying entropies to analyze EEG time series is quite an advantage as has been frequently reported [21,22,23]. In the next section, we describe the innovation of applying our entropy parameters to a set of EEGs collected from stressful situations. Such an EEG database was registered and communicated in [24,25,26] and it is also a novelty as a result of the aforementioned ethical considerations. The experiences and reactions of the participants are parallel to those suffered by attendants. Hence, as a result of the ethical and technical difficulties to create and collect EEG from caregivers, we decided to investigate a database of stressful events that reflect a comparable burden of parents and caretakers.

The structure of the manuscript is as follows. In Section 2, the subjects and the database are described explaining the stressful events. In Section 3, the evolution of entropy parameters that converge to ours is briefly studied. Section 4 provides the outcomes; entropy indices and graphs to associate complexity of time series to stress and brain lobes activity. Section 5 discusses the main results. At the end, conclusions are presented.

Results in terms of times series entropies allow us to offer the following contributions:Traditional EEG analysis must be processed serially, i.e., one at a time. In contrast, entropies enable us to make a set of EEG comparison at the same time.Dealing with long EEG implies numerical drawbacks to other analysis parameters, while our entropy processing does not have that disadvantage.Caregivers have generally received little to no attention. This is the first time a collection of EEG from caregivers has been analyzed by our entropies. As a consequence of (1), we can offer in a single plot contrasting results based upon patient/event, patient/brain zone, or a combination.The latter (3) enables the identification of the worst/best event objectively and immediately.Computation time is certainly not an issue in our dynamic entropies.Graphical results facilitate objective interpretations.

## 2. Subjects and EEG Database

The use of time series entropy in the study of stress from EEG is relatively new [19,27,28]. However, as explained in Section 1, the availability of EEG in caregivers stress is rather rare or absent. Nevertheless, it was possible to find a collection of EEG of this type in [25,26] where anxiety was evaluated and measured. The EEG was related to 23 healthy (This means that the participants do not suffer from anxiety disorders and that they feel they can tolerate the activities) participants (13 women, 10 men, mean age 30) during anxiety elicitation by means of face-to-face psychological stimuli. In an isolated location, the subjects were asked to carry out the experiment individually. Each of them kept eyes closed and minimized gestures and speech in order to register EEG signals with as few artifacts as possible. All the activity was developed in the presence of a psychotherapist who starts reciting the event and assisting the person to imagine it, inducing anxiety. This stage consists of two phases: Recitation by the psychotherapist during the first 15 s and Recall by the subject during the last 15 s. The latter is studied by us by means of dynamic entropy indices (1920 samples).

The recording of data was performed with an Emotiv Epoc that has 14 (+2 references) sensors placed at the following locations: AF3, AF4, F3, F4, FC5, FC6, F7, F8, T7, T8, P7, P8, O1, and O2 (14 channels). The sample frequency is 128 Hz, i.e., ≈7.81 milliseconds per sample. The amount of files is 1920 samples, i.e., 15 s time. In addition, 12 events were handled. They correspond to 12 stimuli grouped in 3 categories (see Table 1). From this, we defined a 4D matrix of the size 23 × 1920 × 14 × 12. Therefore, a collection of 23 × 14 × 12 = 3864 time series composed of 1920 samples were analyzed. The link in [26] provides raw and preprocessed data. We used the previous one in this research to guarantee better performance. Artifacts and noise were removed and a filtering was performed.

Something that draws attention in this database is the three classes of events (questions) posed to the participants. These three groups of queries resemble the Derogatis Stress Profile (DSP) [29], a 77-item self-report inventory derived from interactional stress theory. The latter considers that stress consists of three interactional components, environmental events, personality mediators, and emotional responses, that, in [25,26], coincides with *external, interpersonal, and internal*, respectively. The set of events are described in Table 1. Each activity was called an “event” and it is denoted with an “E” and a number according to a pre-established ordering. Note that, for E12, a question was re-evaluated by the psychotherapist to adjust the participant’s anxiety level [25,26]. The specificity of each stimulus comes from the Hamilton Anxiety Rating Scale [25,26,30].

The way we propose to process all this information is by means of dynamic bivariate multiscale entropy (DBMSE). This parameter of ours has successfully helped to unveil information in epileptic encephalopathies cases [8,9,10,11,12,13,16,31]. Now, DBMSE is applied to an EEG dataset recorded during controlled stressful experiences. Entropy methods are model-less analytical resources to compute the degree of complexity (non-regularity) of data without any assumption about the origin of it. It is worthwhile to mention that entropy methods work in the temporal structure of data regardless of its associated momentum statistics.

## 3. Methods

EEG signals are registered according to the well known 10–20 standard system ([32,33,34] and Figure 1). Each electrode provides a brain record per region. Such a register is a sample voltage vector, i.e., a time series. It can be non-complex or regular (as a periodic signal) or complex (as an EEG). With the purpose of measuring the complexity of a time series, we utilize the concept of *time series entropy* [28,35,36,37]. In information theory, Shannon defined TS-entropy as a measure to calculate how regular a sampled signal is [37]. After that, the idea of using an entropy parameter to physiological signals was conceived by Pincus [28]. There, Approximate Entropy (ApEn) was designed. The name “approximated” arose as a consequence of wishing to numerically determine a mathematical expression in terms of limits (actually the Kolmogorov–Smirnov definition of entropy). ApEn estimates the likelihood that sequences of a fixed pattern length *m* that are closed with respect to a threshold *r* remain close when the pattern length is increased to m+1 in a total of *ℓ* samples.

Since ApEn had some disadvantages, an improved version of it was proposed in [35] and the resulting statistic was called Sample Entropy (SaEn). An extra update of this was given in [38] where the obtained parameter was called Multiscale Entropy (MSE). In [13,16,31], we extend it to a parameter with three-dimensional graphic interpretation, called Bivariate MSE (BMSE). Moreover, a collection of BMSE surfaces define a *complexity film*. Associated to each surface, we also obtained an average BMSE referred to as Dynamic BMSE (DBMSE). This value encompasses long term information in a single number [13]. As our parameters matured from the classical MSE, we now describe how to determine and interpret this MSE.

### 3.1. Calculation of MSE

A single EEG channel is represented by a sampled data vector indexed to integer factors of the sample time. Thus, for a TS-EEG channel, *X*, X=[X(1),X(2),⋯,X(ℓ)] where each entry is measured in millivolts and *ℓ* is the TS length. The basic MSE algorithm sets out to specify a subsample time vector τ, i.e., $\tau =[1,2,\cdots ,\tau_{max}]$. Each entry of τ determines consequent MSE values. Observe that τ = 1 implies to work with the original TS length. The collection of MSE parameters linked to each entry in τ are produced as follows. Fix *m* and the length of the sub vectors of *X*, and then we are compare them and call them *u*. For instance, if m=2, a family of sub vectors u(1),u(2),⋯,u(ℓ−m+1) is constructed. Each of them is defined as u(1)=[X(1),X(2)], u(2)=[X(2),X(3)],…,u(ℓ−m+1)=[X(ℓ−1),X(ℓ)]. With the aim of computing closeness between pairs of sub vectors, a set of distances is determined between them as d(1)=|X(1)−X(3)|, d(2)=|X(2)−X(4)|,…,d(ℓ−m+1)=|X(ℓ−m+1)−X(ℓ−m+3)|. Afterwards, the maximum value between two consecutive entries of vector *d*, as max(d(1),d(2)), max(d(3),d(4)),…,max(d(ℓ−m),d(ℓ−m+1)) is obtained. From this set of maximum values, count those for which max(d(i),d(j))<r, i=1,⋯,ℓ−1, j=1,⋯,ℓ−m+1, where *r* is a threshold. Next, compute the natural logarithm of the accumulated count. Divide this result by the length ℓ−m+1 and name is $\Phi^{m}$. Repeat this for m+1 and call it $\Phi^{m+1}$. Determine finally $MSE=\Phi^{m}-\Phi^{m+1}$. Repeat the aforementioned steps for each entry of vector τ. Hence, a single MSE value for each entry of τ will be produced and a *complexity graph*, in terms of MSE, will be plotted. These steps are given in Algorithm 1.
**Algorithm 1** Multiscale entropy (MSE) computationFor each EEG channel, define a TS as X=[X(1),X(2),⋯,X(ℓ)] where *ℓ* is the number of EEG samples.Fix *m*, the length of compared sub vectorsChoose $\tau_{min}$, Δτ, $\tau_{max}$, the minimum, maximum and increment values, resp., of the subsample vector τ.**for** τ = $\tau_{min}$ to $\tau_{max}$
**step**
Δτ **do** Subsample *X* for $\tau_{min},\tau \left(2\right),\ldots ,\tau_{max}$ Fix *q* sub TS from each EEG channel as $X_{1}=\left[X\left(1\right),X\left(2\right),\cdots ,X\left(\frac{\ell }{q}\right))\right]$,…, $X_{q}=\left[X\left(\ell -q\right),\cdots ,X\left(\ell \right)\right]$. **for** i=1 to q **do**   Determine a variable threshold $r(X_{i})$ for each sub TS, $X_{i}$ according to $r\left(X_{i}\right))=0.2\sigma \left(X_{i}\right)$   Construct a sequence of m-long vectors as z={u(1), u(2), …, u(ℓ−m+1)} as u(i)=[X(i)X(i+1)X(i+1−m)]   From the sequence *z*, calculate the number of u(j),j=N/τ such that its distance $d\left(u\left(i\right),u\left(j\right)\right))=max_{i}\left|u\left(i\right)-u\left(j\right)\right|<r\left(X_{i}\right)$   For i≠j, calculate the number of distances (denoted as #) within threshold r(X(i)) as $C_{i}^{m}\left(r\right)=\frac{\#d(u(i),u(j)))<r}{\ell -m+1}$   Calculate the logarithmic average $\Phi^{m}\left(r\right)=\frac{1}{\ell -m+1}\sum_{i=1}^{\ell -m+1}$ln$\left(C_{i}^{m}\left(r\right)\right)$   For each sub TS vector, compute $S_{i}=\Phi^{m}\left(r\right)-\Phi^{m+1}\left(r\right)$ = $\frac{1}{\ell -m+1}\sum_{i=1}^{\ell -m+1}$ln$\left(\frac{C_{i}^{m}\left(r\right)}{C_{i}^{m+1}\left(r\right)}\right)$   Determine the average of all $S_{i}$ and name it $MSE_{i}$ **end for****end for**Plot τ vs. MSE

As an example of the MSE behavior, observe Figure 2. This signal was collected from a child with epilepsy. A similarity pattern can be noticed in the waveforms from t≈200 s to t≈1200 s. As a result of this, the behavior of the MSE values are smaller than one, indicating regularity (similarity to a periodic signal). A recovery stage emerges in the long term MSE>1 [13,16,28]. In addition, the choice of the scale in MSE is crucial [39,40].

### 3.2. Entropy Algorithms and Entropy Indices for Stress Analysis

**MSE entropy index**. From Algorithm 1, we are computing a collection of numerical values of MSE for each subsample time τ (Figure 2). A global complexity measure from this can be specified via an MSE index $\bar{MSE}$, as:(1)$$\bar{MSE}=\frac{1}{n}\sum_{i=1}^{n}MSE_{i},n=\frac{\ell }{\Delta \tau }$$

For the case of Figure 2, $\bar{MSE}=0.8619<1$ (In Algorithm 1, examine the logarithmic term ln$\left(\frac{C_{i}^{m}\left(r\right)}{C_{i}^{m+1}\left(r\right)}\right)$. If $C_{i}^{m}\left(r\right)\approx C_{i}^{m+1}\left(r\right)$, ln(≈1)≈0= MSE). This number indicates a preponderantly regular pattern in the signal. It coincides with wave form exposed in Figure 2.

**Bivariate MSE (BMSE) graph and BMSE index** Plotting MSE graphs and calculating MSE indices, although useful, may take a long time for very long records. For instance, an epilepsy EEG may last even hours. For those matters, it is useful to establish a *complexity surface*. Such structure comes as a result of calculating a *bivariate MSE (BMSE)*. This surface is defined when a set of MSE curves are obtained from long EEG data and the total duration is divided into a certain number of MSE plots. More concretely, if we divide *ℓ* by $n_{c}$, a desired number of curves, we will obtain $\frac{\ell }{n_{c}}$ samples per MSE component plots. In addition, its associated MSE-index, $\bar{MSE}$ is calculated.

Moreover, as explained above, plotting each MSE curve next to each other along time, will produce a complexity surface with τ as a third axes. Indeed, BMSE=MSE(t,τ) will be composed by such collection of MSE pieces as is explained in Algorithm 2.
**Algorithm 2** Bivariate multiscale entropy (BMSE)Define $\ell ,n_{s}$, length of the desired EEG channel *X* and number of component slices (MSE-curves) of the desired BMSE surface, resp.Do the following for each time factor scale: $\tau =[\tau_{min},\tau_{min}+\Delta \tau ,\tau_{min}+2\Delta \tau ,\ldots ,\tau_{max}]$**for** k=1 to $n_{s}$ **do** Apply **Algorithm 1** to sub vectors $v\left(k\right)=[X\left(\left(k-1\right)\left(\frac{\ell }{n_{s}}\right)\right):X\left(k\frac{\ell }{n_{s}}\right)]$ to obtain an MSE curve for each v(k). Use Equation (Equation 1) to obtain $\bar{MSE(v(k))}$ (for each curve). Use Equation (Equation 1) to define a BMSE index as $\bar{BMSE}$ for $\bar{MSE(v(k))}$.**end for**Construct a BMSE surface with the collection of all MSE curves obtained.

In line with MSE index, a BMSE index for a 3D complexity graph can be defined:(2)$$\bar{BMSE}=\frac{1}{n_{s}}\sum_{i=1}^{n_{s}}{\bar{MSE}}_{i}$$
where $n_{s}$ is the number of slices that constitutes the BMSE plot. Thus, a BMSE structure enables the encompassing of very long databases and improves visualization [8,9,10,13,16].

As an example of the latter, let us see what happens in F7 during E1. First, in Figure 3 (upper panel), the F7-time-voltage plot is displayed. In the lower panel, its corresponding BMSE surface is provided (actually, this surface is made of of 7 MSE curves with $\bar{MSE}=2.1220,2.0959,1.4730,1.4335$, 1.5480,1.8157,1.4646, and $\bar{BMSE}=1.7075$). Observe that in the time–voltage plot, almost in second 9, something happens. This behavior worsens at about the 11th second, where the voltage drops drastically. This means a sudden change in the mind of that person. The entropy surface presents higher entropy values for τ=1,2 but beyond that, for all EEG-time, the surface is rather flat with low complexity values. Thus, in the long term, the tendency in F7 is to have a “quiet” feeling as we can see in the time-voltage graph (voltage amplitude decreasing) and in the BMSE surface (flattening). This also coincides with the fact that the stressful activity is almost over, as can be seen from the time–voltage graph. It should be pointed out that the number of samples in this study is rather short ℓ=1920, in contrast with those collected during a long term epilepsy study (in the order of tens of thousands samples, Figure 4). Indeed, inspect the surface constructed in Figure 3. It is much coarser than those given in Figure 4.

**Dynamic BMSE**. Additionally, instead of giving information on a single surface, it is possible to subdivide time to produce multiple complexity surfaces associated with each time piece. This assemblage can be collected and animated to produce a complexity film. See Figure 4.

In accordance with Equations (Equation 1) and (Equation 2), it is possible to have a DBMSE index as well.
(3)$$\bar{DBMSE}=\frac{1}{n_{F}}\sum_{i=1}^{n_{F}}{\bar{BMSE}}_{i}$$
where $n_{F}$ is the number of 3D frames desired to split the EEG time. Since DBMSE is the parameter that encompasses the most information, we now proceed to show and describe the results obtained from the EEG under stress using this DBMSE index.

## 4. Results

We have demonstrated that dynamic entropy parameters have successfully helped to resume long term EEG information of children with epileptic encephalopathies [8,9,10,11,12,13,16,31]. Nevertheless, we now turn the tables. Caregivers are nothing but human beings that have to deal with a wide range of conditions that produce them anxiety and stress. As far as we are concerned, there are no EEG databases that specifically reflect the stress of primary care providers.

We apply DBMSE to assess stress in the 23 subjects through their EEG-database. From this, we produced two sets of curves:A DBMSE index was obtained for each subject per EEG channel, indicating the most affected areas of the brain during such situations.The second one presents a collection of entropy-event curves that shows the behavior of this dynamic parameter for each event and for every participant.

From above, conclusions are deduced to explain the participants’ alterations related to their brain areas per event for every participant.

### 4.1. Entropy by EEG Channel for All Events and All Subjects

As described, the EEG database consists of 23 participants during 11 events with 14 channels. Let us observe the DBMSE index behavior of the first nine persons in Figure 5. There, we first note that the signal is complex, MSE>1. Remarkable is that, for subject 1, complexity of brain activity is almost constant; this means that for this person there was a varied activity in almost all channels. Stress affects almost all senses in this person, paying constant attention to every piece of information received. However, participants 3, 4, 7, and 9 developed less activity in the frontal part. The frontal lobe is in charge of managing higher level executive functions. These are planning, organizing, self-monitoring, and self controlling. Since the frontal region revealed regularity patterns, altered attention and control are manifested by these subjects. A different effect is exhibited by participants 2, 5, and 7 with other combined repercussions. Person 8 presents high/low levels across all brain sections, meaning that he/she is alternating complexity. We consider that this subject is more nervous than the other. This is in line with his/her $\bar{DBMSE}=1.5853$, which is the minimum of the set. The behavior for the other participants can be similarly analyzed. Nevertheless, in Figure 6, we will scrutinize a graph with an overall behavior of all subjects at the same time.

The curves displayed in Figure 6 are the encompassed behavior of the 23 subjects for the 14 channels and all the events. The upper panel shows the $\bar{DBMSE}$ and the lower panel its corresponding coefficient of variation, cv=σ/μ. In that section, channels T7, T8, P7, and P8 are the most active. It can be recognized that the incremental complexity goes from the frontal lobe to the occipital region (see also Figure 1). This curve says that, as an average (per channel), the information processed by the frontal lobe is less relevant than the one handled by the temporal, parietal, and occipital zones. Since the frontal lobe is in charge of planning, organizing, and self-monitoring, people are less stressed under stimuli of these kinds than those processed by the other parts of the brain. The temporal region transforms auditory data linked to encoding of memory. It is also responsible for calculating affect/emotions and some visual facets. The parietal lobe is in charge of touch perception, manipulation of objects, and movement control. Consequently, since the temporal and parietal zones show an incremental rise in entropy (a rise in this type of attention), affect/emotion reactions linked to imagining action (manipulation to avoid the undesirable events) is what this part of the plot suggests. At the end of the curve, occipital activity as visual perception of color, forms, and motion are less excited as a result of the events that imply less processing information of this class; thus, the brain activity increases.

The lower panel exhibits variation among all channels for all participants in terms of the coefficient of variation of DBMSE. By way of illustration, observe the value of cv(DBMSE) at T8 (the lowest amplitude in this curve). This means that T8 has a low variability from one participant to the other; i.e., there was a general trend to have high $\bar{DBMSE}$ values at T8 in all participants for all events in general.

### 4.2. Entropy by Event for All Channels and All Subjects

Now the complement is presented next. Consider Figure 7 where $\bar{DBMSE}$ is obtained per event for participants 10–18. Consider for instance the representative curve of subject 17, where a drastic change occurs in $\bar{DBMSE}$ from E9 to E10. Recalling Table 1, where E9 is defined as *Recalling a bad memory* and E10 as *Fear of getting sick and missing an important event* it can be noticed that $\bar{DBMSE}<1.6$ for E9 and $\bar{DBMSE}>2$ for E10. This means that a higher complexity is associated to E10 which in this case would be obvious. However, the behavior inverts for subject 11. Cases of participants 14 and 15 are similar for these events. As a complement, Figure 8 shows that E2 and E10 evince the highest complex behavior. E2 is *Witnessing a deadly accident Familial instability / Financial instability / Maltreatment* and E10 is *Fear of getting sick and missing on an important event*. After them, E3 (*Witnessing a deadly accident*) and E9 (*Recalling a bad memory*) were those showing more activity. Almost at the same entropy value is E11, *Fear of being diagnosed with a serious illness*. In the lower panel, the coefficient of variation indicates that the event which shows more instability is E7 (*Fear of getting cheated on/Fear of losing someone close*), followed by E11, i.e., there were persons who agreed and did not agree. Alternatively, the lowest variations were perceived in E10, E4, and E6. Roughly speaking, the first events are more stressful than the middle ones but the final questions were, again, the causes of tension.

All the aforementioned conclusions are the average trend per channel and per event for all subjects. Nevertheless, we can also obtain information from each subject’s DBMSE curve (in the sense of Algorithm 1). Placing side by side all the DBMSE curves from each participant (Figure 5 and Figure 7), we can construct another surface (Figure 9). In this illustration, the right upper panel shows the surface as a function of the events and the EEG channels. This structure consists of the 23 DBMSE curves, already described. At the end of the surface, the last participant’s DBMSE curve is highlighted in red and, next to it, the corresponding curve alone. In the left lower section, the surface is projected on the events–channels plane. The red zones represents holes in the surface and are low entropy regions. These zones correspond to the events and channels given in the table next to the graphs. From this, note that E5, E6, E7, and E8 characterize the repetitive trend in the mind; the interest is lower than in other events (see Table 1). This deduction is in line with the Figure 8. Now notice that E6–E7, E6, E8, E5 activates F7, FC5, F3, and AF4, respectively, and this matches the Figure 6. Additionally, observe the curve of the coefficient of variation in each of these cases where it also coincides. Recall that entropies calculated for each time-lag are an estimated of regularity, i.e., predictability of the EEG channels. These entropy parameters assess the impact of the brain dynamics from an inter-lag viewpoint. As already explained, higher entropy quantities indicate unpredictability of the brain region (channels). People are less worried about E5–E8 than about other events, but not all think the same (coefficient of variation). In addition, notice the blue zones of the projected surface. These regions roughly cover E1–E2 and E9–E12, that are events with more importance.

## 5. Discussion

It has been explained that in order to obtain the entropy values and plots given in Section 4, a set of 3864 time series were processed. DBMSE in terms of numbers and graphs, encompass all the input TS information. The main result can be stated as follows. Events E2 and E10 triggering frontal and temporal channels revealed the actual state of participants under stressful situations. Subject 1 proved to be the most stressed. Knowing all these types of deductions and extrapolating these results to actual caregivers for those with epilepsy or with other serious affections will help to find solutions to mitigate undesired and damaging conditions that might produce anxiety disorders with physiological consequences. Even in a controlled setting, the impact of all this is really relevant because a direct projection about real caregivers is possible with real repercussions in health. At this point we go back to the reflection performed in Section 1 where ethical commitments versus necessity of real EEG resources in stress and anxiety affections should be revised.

Furthermore, from the original 12 × 14 × 23 = 3864 times series processed, the DBMSE concept permitted us to produced 23 plots composed of 14 DBMSE points (channels) each, plus 23 graphs comprised of 12 DBMSE points each (see Figure 5 and Figure 7), yielding 23 + 23 = 46 curves per subject. The latter are accompanied by an average curve and its corresponding coefficient of variation plot, completing so a total of 46 + 2 + 2 = 50 plots (25 from the channels analysis and 25 from the events information). This is most compacted way to display a huge amount of information via DBMSE from a lot of EEG-times series [8,9,10,11,12,13,16,31].

On the other hand, we are aware that if more data were available for caregivers in real stress situations it would enrich our study. As a result of that, we are cognizant that, although useful, our investigation can be categorized as a pilot study. Recall that a pilot sudy is defined as an investigation that describes a fundamental phase of the research process with the purpose of examining the feasibility of an approach that is intended to be used at a subsequent larger scale [41,42].

It is noteworthy mention that there do exist international organizations in favor of caregivers. However, although really relevant, this help is social and psychological. By way of illustration, in [43], people can find a “Caregiver Support Group That’s Right for You”. Assistance is very varied and associations help when caregivers deal with Alzheimer’s and dementia, cancer, heart disease and stroke, etc. Particularly noticeable is [44] where caregiver’s support helps in cases of a very difficult-to-treat childhood epileptic encephalopathy; i.e., Doose syndrome (DS) or Lennox–Gastaut syndrome (LGS) [32,33,34,45].

Becoming a caregiver for a child with serious medical disorders can be both immensely gratifying but hugely challenging at the same time. Frustration, segregation, medical assistance, treatment, and support services can be overwhelming for parents, particularly when they are the main custodians.

As mentioned in the Abstract, the advantages of our entropy measure permitted us to conclude about which event and brain region are the most affected. The main benefits of our study are already given in Section 1. However, some drawbacks are:Short term databases may be suitable of miss-processing. However, according to literature, the databases worked out here are in the limit of such length.The EEG we worked with comes from third party and we must trust that any preprocessing was correctly completed.Shortage or nullity of available EEG of caregivers limit us of further comparisons.

## 6. Conclusions

Databases of 23 participants were analyzed via DBMSE. Each subject has a 14 channels × 12 events-EEG records (3864 time series consisting of 1920 samples, i.e., 15 s). The results showed that EEG channels associated to the frontal and temporal lobes were the most stimulated by stressful activities. Complementing the latter, events E2 and E10 were the most stressful by connection with high entropy values. Even in this controlled environment, the conclusions were in line with the expectation and can be extrapolated to real caregivers in serious diseases as childhood epilepsy.

## Figures and Tables

**Figure 1 ijerph-20-05913-f001:**
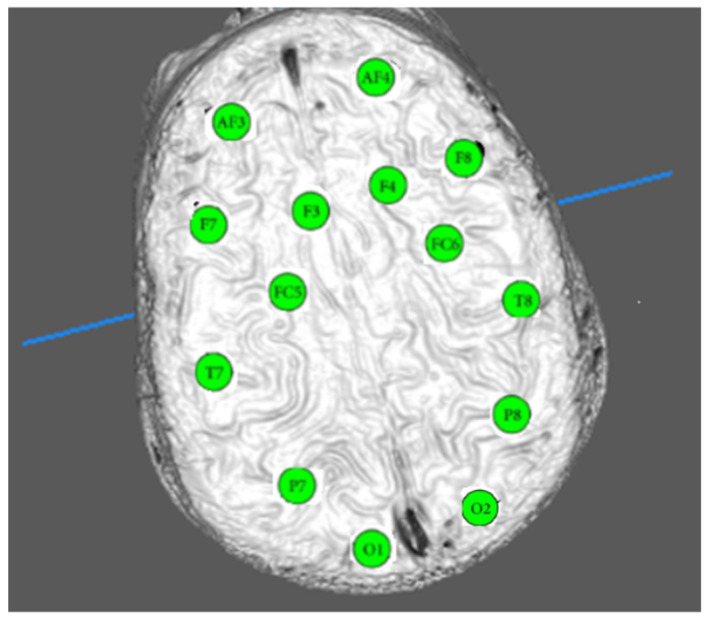
The 14 channels monitored to obtain DBMSE graphics.

**Figure 2 ijerph-20-05913-f002:**
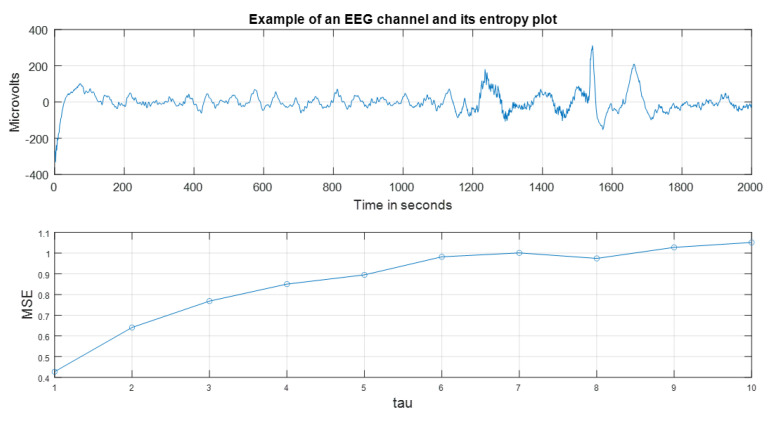
MSE in F3 in case of infantile seizure. Observe the similarity patterns in the time signal and the low values of MSE, i.e., MSE <1 for τ<7, [13,16,28].

**Figure 3 ijerph-20-05913-f003:**
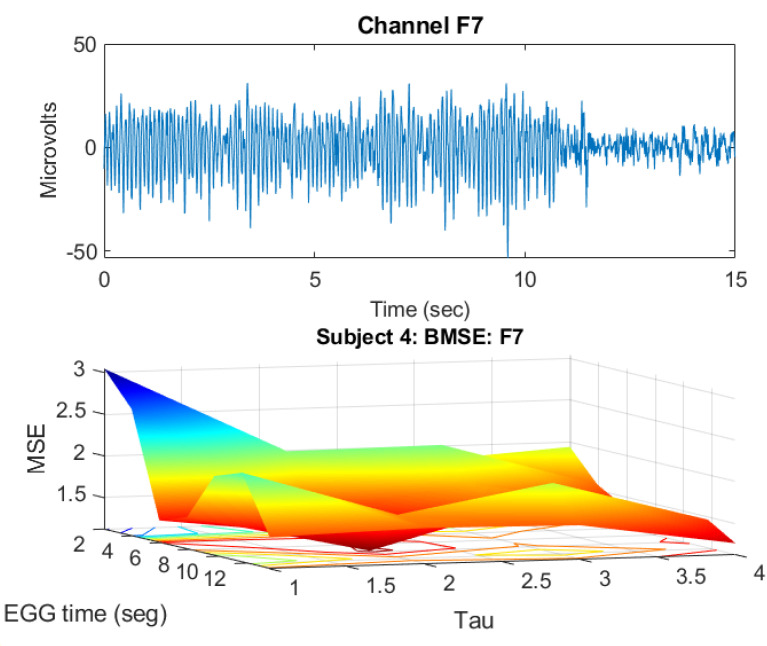
Subject 4, channel F7. Time-voltage plot and its corresponding BMSE surface.

**Figure 4 ijerph-20-05913-f004:**
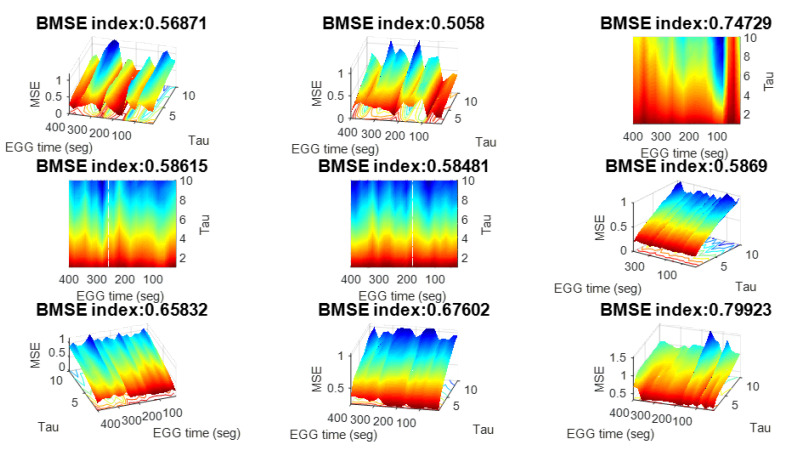
Example of consecutive BMSE surfaces and associated BMSE indices in a case of epilepsy. Blue zones denote high entropy activity (no seizures) and red/orange tones indicate seizures. BMSE indices are also called DBMSE indices according to Equation (Equation 3). Notice that this collection of surfaces can define a *complexity movie* [9,13].

**Figure 5 ijerph-20-05913-f005:**
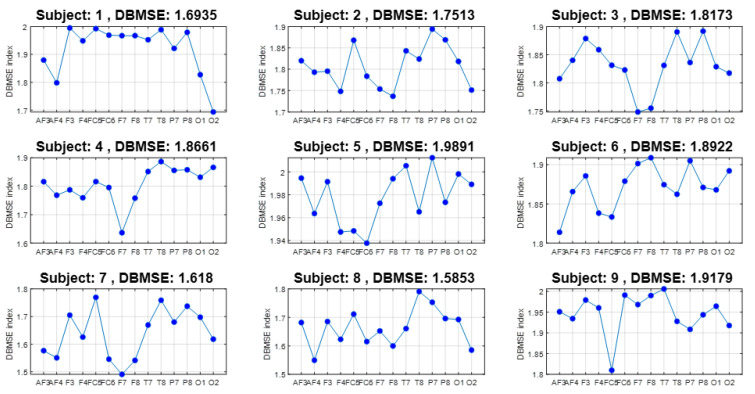
DBMSE per channel for subjects 1–9. With respect to brain lobes, subject 1 is the most stressed as it has the highest entropy values for almost all channels, thus, reflecting a constant alert state.

**Figure 6 ijerph-20-05913-f006:**
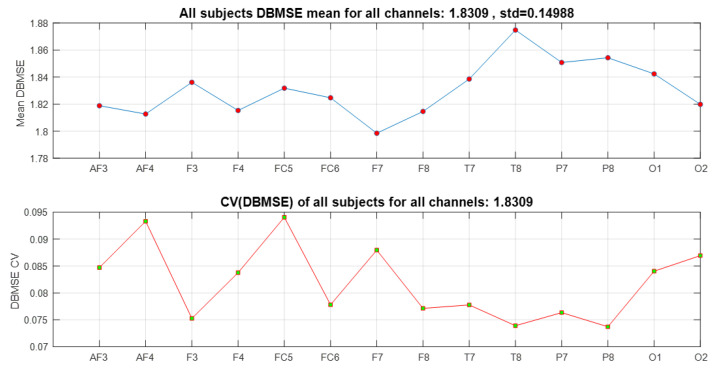
Mean DBMSE, per channel, for the 23 subjects. The active brain areas in all subjects (averaged) are T (temporal), P (parietal), O (occipital), and F (frontal). This is in line with the fact that T encodes emotional information and P processes auditory and visual data.

**Figure 7 ijerph-20-05913-f007:**
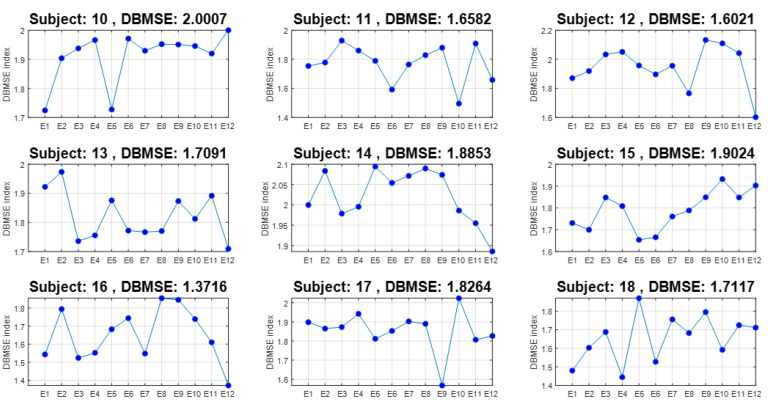
DBMSE for subjects 10–18 for all events.

**Figure 8 ijerph-20-05913-f008:**
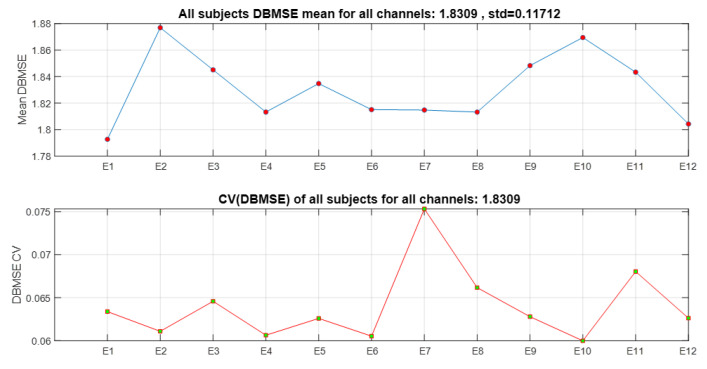
Mean DBMSE for all events for all subjects.

**Figure 9 ijerph-20-05913-f009:**
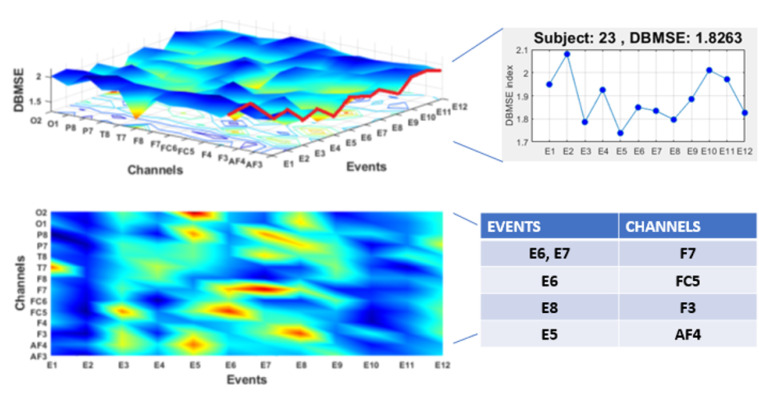
DBMSE in terms of brain zones and events. Crucial values of entropy are colored in red and correspond to determined events and channels. The red plot at the border of the surface corresponds to the curve given in the right upper part. Thus, a 3D surface is a general case for the (projected) curve just described.

**Table 1 ijerph-20-05913-t001:** Grouping of stimuli applied to participants and corresponding events labeling.

CATEGORY	STIMULI	EVENT
EXTERNAL	Witnessing a deadly accident.	E1
	Fam./Fin. instability/Maltreatment.	E2
	Deadlines/Insecurity/Routine.	E3
INTERPERSONAL	Relationship with the supervisor.	E4
	Lack of confidence towards spouse.	E5
	Being in an embarrassing situation.	E6
INTERNAL	Fear of getting cheated/losing someone.	E7
	Fear of children’s failure/Feeling guilty.	E8
	Recalling a bad memory.	E9
	Fear of disease and missing an event.	E10
	Fear of suffering a serious illness.	E11
	Re-evaluation of some items.	E12

## Data Availability

As explained in Section 2, the database of subjects under stressful stimuli are available in https://doi.org/10.21227/barx-we60. Our results processing that file are available on request.

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
