# Peer review of "Stress Response Analysis via Dynamic Entropy in EEG: Caregivers in View"

_ijerph, 2023, doi:10.3390/ijerph20105913_

Round 1

Reviewer 1 Report

This study aimed to propose an approach to evaluate stress response using eeg signals. I have the following suggestions.

What is the novelty of this study although several approaches to evaluate stress response using eeg signals have been proposed earlier?

Please write down the contribution of the study at the end part of the Introduction section in bulleted form.

The abstract should be improved by combining the objectives, short methodology, main findings results, and prospective application.

EEG is highly sensitive to the powerline, muscular, and cardiac artifacts. In EEG data preprocessing, authors need to mention how you handle AC power, ECG, and EMG artifacts in EEG signals. Do the authors think that their proposed method is robust to such kinds of artifacts?

Authors should introduce the EEG applications in ML/DL-based disease, and mental workload prediction in broad scope, such as article, Explainable Artificial Intelligence Model for Stroke Prediction Using EEG Signal; in article, healthsos: real-time health monitoring system for stroke prognostics; in article, quantitative evaluation of task-induced neurological outcome after stroke; in article, driving-induced neurological biomarkers in an advanced driver-assistance system; and in article, quantitative evaluation of eeg-biomarkers for prediction of sleep stages.

No clear research outcomes not reported in the result section. Authors should report the entropy changes with varied levels of stress.

Title should be revised. Caption of Table 1 is inappropriate.

Authors should report more performance measures of entropy in response to stress.

The discussion section needs to be improved. Authors must make discussion on the advantages and drawbacks of their proposed method with other studies adding a table in the discussion section.

Clinical explanation of these findings needs to be described in support of reference.

From the writing point of view, the manuscript must be checked for typos and the grammatical issues should be improved.

Author Response

Dear reviewers,

Thank you very much for your valuable comments. We have uploaded a Word document with your remarks taken into account. 

Reviewer 2 Report

The paper Stress Response Analysis via Dynamic Entropy in EEG: Caregivers in View said they propose a novelty time series (TS) entropies to inspect collections of EEGs got during stress situations. But in my opinion, this method is the improvement of MSE at most. In general, the experimental design in this paper is novel and meaningful. But the method is not a new one, and there are some flaws. The following are my detailed comments:

1.    In the line 197, for the case of figure 2, the mean of MSE is 0.8619, why you compare it with 1? This is groundless, it is not proved that the maximum local mean of MSE is equal to 1. Actually, the value of MSE is able to be larger than 1, as well as the mean of MSE.

2. The definition of BMSE is lack of physical significance. Because, the tau axis is the coarse-grained time series, it is a scale of time. And the time seg is still a variate of time. The MSE graph meaningless.

3. DBMSE is the further average of BMSE, does not reflect the dynamic variability of the signal.

4. In fact, the select of scale in MSE is crucial. As the frequency band of EEG is various, and represent different physiological meanings. In reference to [1] A comparison study on stages of sleep: Quantifying multiscale complexity using higher moments on coarse-graining, Communication in Nonlinear Science and Numerical Simulation, 2017, 44: 292-303. [2] Changes in EEG multiscale entropy and power-law frequency scaling during the human sleep cycle. Human Brain Mapping, 2019, 40,2, 538-551.

5. In this manuscript, I see the scale is less than 10, this is far less than the scale that can be discussed. The sample frequency is 128 Hz in this study, that is to say the frequency band you discussed just cover the β band, the large scales, α,θandδ are dismissed. This is incomplete when discussing the physiological significance of stress.

6.  Lack of Simulations.

In my opinion, the manuscript in the current version cannot be accepted by this high-level journal. 

Sincerely

Author Response

Dear reviewers,

Thank you very much for your remarks. We have uploaded a Word document with your comments taken into account. 

Round 2

Reviewer 2 Report

I have no other suggestions.